# Current Development and Future Perspective on Natural Jute Fibers and Their Biocomposites

**DOI:** 10.3390/polym14071445

**Published:** 2022-04-01

**Authors:** Sweety Shahinur, M. M. Alamgir Sayeed, Mahbub Hasan, Abu Sadat Muhammad Sayem, Julfikar Haider, Sharifu Ura

**Affiliations:** 1Department of Testing and Standardization, Bangladesh Jute Research Institute, Manik Mia Avenue, Dhaka 1207, Bangladesh; 2Dyeing and Printing Division, Bangladesh Jute Research Institute, Manik Mia Avenue, Dhaka 1207, Bangladesh; mmasayeed70@gmail.com; 3Department of Materials and Metallurgical Engineering, Bangladesh University of Engineering and Technology, Dhaka 1000, Bangladesh; mahbubh@mme.buet.ac.bd; 4Fashion Institute, Manchester Metropolitan University, Cavendish Street, Manchester M15 6BG, UK; asm.sayem@mmu.ac.uk; 5Department of Engineering, Manchester Metropolitan University, Chester Street, Manchester M1 5GD, UK; 6Division of Mechanical and Electrical Engineering, Kitami Institute of Technology, 165 Koen-cho, Kitami 090-8507, Hokkaido, Japan; ullah@mail.kitami-it.ac.jp

**Keywords:** natural fiber, jute fiber, nonwoven, woven fabrics, polymer composite, thermoset plastic, thermoplastics

## Abstract

The increasing trend of the use of synthetic products may result in an increased level of pollution affecting both the environment and living organisms. Therefore, from the sustainability point of view, natural, renewable and biodegradable materials are urgently needed to replace environmentally harmful synthetic materials. Jute, one of the natural fibers, plays a vital role in developing composite materials that showed potential in a variety of applications such as household, automotive and medical appliances. This paper first reviews the characterization and performance of jute fibers. Subsequently, the main focus is shifted towards research advancements in enhancing physical, mechanical, thermal and tribological properties of the polymeric materials (i.e., synthetic or biobased and thermoplastic or thermoset plastic) reinforced with jute fibers in a variety of forms such as particle, short fiber or woven fabric. It is understood that the physio-mechanical properties of jute-polymer composites largely vary based on the fiber processing and treatment, fiber shape and/or size, fabrication processes, fiber volume fraction, layering sequence within the matrix, interaction of the fiber with the matrix and the matrix materials used. Furthermore, the emerging research on jute fiber, such as nanomaterials from jute, bioplastic packaging, heavy metal absorption, electronics, energy device or medical applications and development of jute fiber composites with 3D printing, is explored. Finally, the key challenges for jute and its derivative products in gaining commercial successes have been highlighted and potential future directions are discussed.

## 1. Introduction

Due to their low carbon footprint, natural fiber-based products are attracting significant attention among academic and industrial researchers for sustainable product development. Amongst the commonly used natural fibers, more recently, a renewed research interest was noticed in the published scientific work on jute plant cultivation (such as diseases and pest identification and control, plant varieties development, seed and plant entomology variation observation at different conditions, genome sequence, soil and fertilizer management and retting process) as well as multidimensional applications of jute fiber in product development including fiber-reinforced polymer composite, micro-crystalline cellulose (MCC), nano cellulose, activated carbon, jute pulp production, 3D printing using the jute fiber as shown in Figure 1. The growing interest in jute fibers can be well attributed to all three aspects of sustainability, that is, environmental, economic and social.

Jute plants grow in grassy soil, requiring 125–150 mm of rainfall per month, mild to moderate temperatures (20–40 °C) and high relative humidity (70–80%) for the best growth. Jute is a bast fiber and grown in plantations, where it is collected once the plant matures. The plants are then typically retted in slow running water so that the fibers can be more easily extracted. Other processes to extract the fibers without retting can be used if required. The jute fibers are then dried and marketed for further processing [1,2]. Yarn is often produced from jute fibers and jute fabrics are produced from jute yarn. Some facts and figures about the jute fiber are shown in Figure 2. In some cases, jute is treated [3,4] via different media to enhance its functional properties. Furthermore, jute is a sustainable raw material for conventional textile products (hessian, carpet backing cloth (CBC), sacking, shopping bag, geo-textile and nursery pot), chemical products (namely carboxy methyl cellulose (CMC), microcrystalline cellulose (MCC) and pulp) and composite products for structural applications (beam, plate, bar). From the sustainability and mechanical properties points of view, jute fiber is better than many natural or synthetic fibers [5]. The development of structural material in the form of thermosets, as well as thermoplastic-based fiber-reinforced composites, has attracted the most attention. The incorporation of jute fibers in a synthetic polymer matrix could improve its strength but at the same time contribute to environmental sustainability through reduced use of synthetic polymer materials.

Even though significant research work has been carried out on jute fibers and relevant review papers have been published [6,8,10,11,12,13,14,15,16,17,18,19,20,21,22], they were dedicated to jute as a sustainable raw material for mainly composite material development without detailing much about the recent advances in other jute-based research such as nanoparticles from jute, jute fiber engineering by nanoparticles or modern 3D printing of jute-based composites. This review starts with summarizing the characteristics and performance of the jute fiber and functional fiber treatment (Section 2). Section 3 describes the fabrication process of the jute reinforced composites and their characteristics. Section 4 highlights the future potential of jute fiber in terms of research and product development. Challenges and potential future research directions are discussed in Section 5. Finally, Section 6 draws a few conclusions based on the review. It is believed that this review would provide useful guidance on the current development and future potential of jute-based products.

## 2. Jute Fiber Characterization

Like any other natural fiber, the performance of the jute fiber varies due to the natural variability in surface and internal microstructural characteristics, which can be influenced by a number of factors including growing conditions (i.e., temperature, humidity, soil condition), retting (water, dew or enzymatic) and fiber extraction processes, fiber length and diameter, chemical constituents and their proportional amounts [6,12,23]. Microstructural characteristics again contribute to the physical, mechanical and thermal performance of the fiber. This section starts with some general morphology of the jute fiber followed by examples of its performance in the form of fiber, yarn and fabric either woven or nonwoven. Finally, examples of functional treatment of the jute fiber in order to improve its performance for a specific application are discussed.

### 2.1. Morphology and Microstructure of Jute Fiber

The fiber morphology is characterized by a scanning electron microscope (SEM) after applying a gold coating on a small portion of the fiber. The related SEM fractured and fiber surfaces are shown in Figure 3. The cross-sectional geometry is characterized by an irregular shape. Furthermore, the cross-section size varies along the length of the fiber [24]. The surface morphology is generally rougher in nature with evidence of random micro pores. The surface morphology of different portions of the jute fiber was found dissimilar due to their variations in diameter and maturity (immature, fully mature and overmature) [9]. Less micropores in the surface morphology generally indicate immature fiber. Further details about the jute morphology can be found in the literature [25,26,27].

Figure 4 presents X-Ray diffraction (XRD) and Fourier transforms infrared (FTIR) spectroscopy spectra of raw jute fiber. The distinctive large XRD peak indicated the cellulose crystallographic plane (002) with a 92.02% degree of crystallinity [28]. FTIR spectra confirmed the presence of functional groups on the jute fiber surface. Several broad peaks across the spectrum indicated the presence of hydrate and hydroxyl (OH), which represented cellulose and hemicellulose elements in the jute fiber [4].

### 2.2. Jute Fiber/Yarn/Fabric Performance

A tensile test is carried out to evaluate the performance of jute fiber/yarn/fabric. Some of the test results available in the literature are shown in Table 1.

#### 2.2.1. Single Jute Fiber

Although, in practical applications, jute fibers are agglomerated as bundles, for standardization, evaluation of single fiber properties is important. Shahinur et al. [9] conducted single fiber tensile testing using an Instron™ universal testing machine (Norwood, MA, USA) by varying the span length (5 mm, 15 mm, 25 mm and 35 mm) with a crosshead speed and load cell of 5 mm/min and 5N respectively. The fibers were then stacked between two paper frames [29] to conform good gripping to the clamps of the test machine and to provide straight direction during the test as shown in Figure 5. Tensile strength, Young’s modulus and strain to failure were determined from the test. The whole jute fiber was cut into three portions—top, middle and cutting/bottom. With an increase in the span length, tensile strength decreased, tensile modulus increased and strain to failure decreased. Furthermore, tensile properties of the middle portion of the jute fiber displayed better performance compared to the top and bottom portions. Although the tensile strength of the jute fiber varies between 200–800 MPa, there are other manmade fibers such as glass fiber (2400 MPa) or more recently developed Forcespinning^®^ NCST fiber showed record-high tensile strength (13.7 ± 3.2 GPa) [36].

#### 2.2.2. Jute Yarn

As jute is a natural fiber, its mechanical properties would vary, indicating that uncertainty exists in obtaining fibers with consistent mechanical properties. The way mechanical tests are conducted on jute might influence the distribution of its mechanical properties. A number of studies were reported in the literature to understand these variations with statistical distribution.

For example, samples were prepared by varying gauge length of the jute yarns samples and stress, strain at failure and Young’s modulus were tested under static tension [37] according to the standard ASTM D578 using 50 mm gauge length. Wide dispersion in mechanical properties of the jute yarn was found and this dispersion becomes steady after testing 200 samples. The experimental data matched well with the data obtained from a two-parameter Weibull analysis.

Smail, et al. studied the mechanical properties of jute yarn at different loading rates (0.3, 1.2, 2.0 and 4.0 mm/min) under static tensile test conditions [38]. The results indicated that loading rate affected stress, elongation and Young’s modulus without any clear trend. However, the best mechanical properties were attained at a loading rate of 2 mm/min.

Ullah et al. [2,39] employed statistical analyses and possibility distributions to ascertain the uncertainty associated with mechanical properties (tensile strength, modulus of elasticity and strain to failure) of jute yarns and fibers by conducting tensile tests. Typically, the jute yarns showed linear behavior until failure in a load versus elongation graph as shown in Figure 6. Although, in another study, this behavior was stepwise due to the gradual failure of the weaker fibers in the yarn during the tensile testing [37]. The uncertainty associated with the jute fibers was found higher than that of the jute yarn. Therefore, it was recommended to use the mechanical properties related to the jute yarn in order to develop jute-fiber-based products.

#### 2.2.3. Jute Woven/Nonwoven Fabric

Jute fabrics come in different weave structures, of which 1/1 plain weave is the most common and such fabric is commercially known as hessian (Figure 7), whose specification is presented in Table 2. In practice, the structure and weight of jute fabric are determined using the standards BS EN 1049-2:1994 and BS 2471:2005, respectively. The breaking force and elongation of the fabric were analyzed following the test standard BS ENISO 13934 and using a tensile tester (e.g., Testometric Micro 500, Rochdale, UK). The breaking forces in the wrap direction were higher than in the weft direction due to the higher number of ends per unit length [40,41]. Minor differences also existed in different batches of fabrics possibly due to the variations in manufacturing or fiber quality.

Meanwhile, jute nonwoven fabrics are generally made from raw jute fiber or jute waste using the needle-punching process. The performance of the nonwoven fabric is evaluated by tensile strength, bursting strength, compressibility, bending modulus, air permeability. Further details can be found in a review on nonwoven fabric [42].

### 2.3. Functional Treatment of Jute Fiber

Being a natural fiber, jute possesses several weaknesses such as easily rotten, flammable, thermal degradation and high affinity to moisture, which make it unsuitable in raw condition for functional use in products. Therefore, they require additional modification for use in sustainable product development [4,28,43,44]. Several modifications have been suggested in the literature to improve the functional performance of the jute fiber alone or jute as a reinforcement in composite materials (Figure 8).

Jute fibers were treated with rot- (RT), fire- (FT) and water-retardant (WT) chemicals and FTIR analysis evidenced the presence of constituents related to the treatment chemicals. Higher bacteria inhibition zone diameters against several bacteria such as *Acinetobacter* sp., *Bacillus cereus* and *Pseudomonas* sp. indicated the effectiveness of rot retardancy treatment. On the other hand, fire- and water-retardant treatments slowed down the burning rate against fire and reduced contact angles in contact with water when compared with the untreated jute fibers. Thermal characterization of the RT, FT and WT jute fibers was conducted using Thermo-gravimetric analysis (TGA) and differential scanning calorimetric (DSC) analysis [44]. Lower heat absorption was observed by the treated fibers when compared to the raw jute fiber. The ranking of the treated fibers in terms of heat absorption (lowest to the highest) was as follows: Raw jute < RT < WT < FT. This indicated that all the treated fibers showed better thermal characteristics than the raw jute fiber, but the FT treated fiber showed the best thermal characteristics.

Natural fiber has high affection for the water but lower affinity to the polymer matrix materials and this drawback can be minimized using chemical treatment and incorporating the treated fiber into the polymer composite. Chemical modification techniques such as alkali or coupling agents like silane are most commonly applied to improve interfacial bond strength in the composites by reducing the hydrophilic nature of the jute fiber to make it compatible with hydrophobic polymer matrix, encouraging chemical bonding and increasing fiber surface roughness to promote mechanical interlocking. These changes make the resulting composite with improved mechanical properties [45]. However, other physical techniques such as plasma or Gama radiation are effective in improving mechanical properties. Furthermore, jute fiber treatment might help in accommodating higher fiber volume in the composites in order to make them lighter and more sustainable. Jute fiber treatment in itself is a huge topic for review, which is outside the scope of this paper. However, further details about different types of jute fiber surface treatment are summarized in recent publications [7,25] and other reviews on plant fiber treatments [46,47,48]. When treated and non-treated jute fibers are incorporated into the matrix, the performance of the composite might change.

Other than changing the chemical structure of the jute fiber to a favorable state by surface treatment, physical and mechanical properties might be affected. It was reported that alkali treatment of jute fiber could reduce its diameter by 8% whereas the tensile strength could be increased by 44% [49].

## 3. Jute Based Composites: Processing and Performance

Natural fibers are incorporated into polymeric materials to form composites with improved and customized properties. A multitude of parameters related to the type of fiber used, pre-processing of fiber, composite manufacturing techniques and internal condition of the composite can affect the performance of the composites as shown in Figure 9. In this section, a brief overview of the fiber forms used in different matrix materials is provided. Performance evaluations of both the thermosetting and thermoplastic composites were reviewed with examples of different forms of fibers. Finally, the progress on the hybrid jute composite development is briefly discussed.

### 3.1. Matrixes for Jute Fiber Reinforced Composite

The mechanical performance of the composite materials depends on the polymer matrix materials such as thermoplastics and thermoset or biobased plastics. Properties of some commonly used matrix materials are presented in Table 3.

#### 3.1.1. Thermoset Matrix

Thermoset matrix materials such as epoxy, phenolics, polyester, bakelite, melamine formaldehyde, and polyurethane come in liquid form at room temperature, and they are mainly sourced from non-renewable resources. Once the matrix material is fully cured, it cannot be reheated, recycled or reused nor can it be biodegraded. They can be processed by hand lay-up, spray-up, vacuum infusion, vacuum bagging, prepag, resin transfer molding (RTM), filament winding, pultrusion, extrusion, resin impregnation and cold press as shown in Figure 10. However, endeavors are made to develop bio-based thermosets from vegetable oils, natural phenolic complexes and other sources, which might possess better thermal stability compared to thermoplastic [52]. Some progress has also been made toward developing biodegradable thermoset plastics with similar characteristics as the traditional thermoset such glycix, titan and hydro [53]. Mass scale manufacturing techniques such as plastic molding of biodegradable thermosets are also being explored in order to get a sustainable thermoset, which can contribute toward reducing carbon footprint.

#### 3.1.2. Thermoplastic Matrix

Thermoplastics can be available in the form of granules, plates, sheets or powder. Polyethylene (LDPE/HDPE), polylactic acid (PLA), polyvinylchloride (PVC), polystyrene, polypropylene (PP), Nylon and polycarbonate are commonly used matrix materials (Figure 11). These types of plastics are recyclable, some of them are biodegradable and as well processed from biobased resources such as PLA. The thermoplastics are processed by compression molding, extrusion, injection molding and sheet molding.

### 3.2. Jute Fiber as Reinforcement in Composites

Jute fibers in the form of particles (macro, micro or nano), fiber (short < 5 mm; semi long > 5 mm; long), fabrics (Plaine, twill and Satin) and nonwoven can be used to manufacture composites depending on the applications and performance requirements. Figure 12 presents different forms of the jute fiber, that will influence the properties of the resulting composite material.

When jute fiber is incorporated into the matrix material, the resulting composite strength would be higher than the matrix material but lower than the jute fiber as shown in Figure 13 [41,43]. Moreover, in the case of thermoplastic composite, there is no crosslinking in the matrix whereas only Vander walls force is present. However, crosslinking is available in the thermoset polymer material itself. Therefore, there is a low possibility of the presence of crosslinking between the fiber and matrix for thermoset composites and a higher possibility in the case of thermoplastic composites.

### 3.3. Processing of Jute Fiber Composite

A number of techniques are used for fabricating jute reinforced composite including molding techniques such as compression molding, injection molding, resin transfer molding, extrusion, pultrusion, hand lay-up and spray lay-up. Processing techniques are mainly selected depending on the type of polymer, requirement of labor, mold condition (open or closed), the form of fiber (short, long, particle), presence of multiple fibers, quality of the end product, speed and cost of processing, capital investment and so on [13,50,54,55]. The popular techniques for fabricating thermoplastics and thermoset plastic composite are hot compression and hand lay-up, respectively, as shown in Figure 14. The details of the jute composite processing techniques have already been reviewed in a recent paper [18]. As the focus of this paper is more on new areas of jute research, interested readers are directed to the article for further details.

### 3.4. Jute Thermoset Composite

Thermoset polymer has high tensile properties particularly tensile strength and young’s modulus whereas poor ductility. By incorporating natural fiber, the strength, as well as ductility, can be further improved. Meanwhile, the thermoset materials are cheap, easily available and, after curing, they become harder at room temperature. In addition, the thermoset plastics are amorphous whereas, after the incorporation of the fiber, the composite possesses both amorphous and semi-crystalline phases [3], which lead to higher strength and strain to failure. In general, nonwoven and woven jute fibers are incorporated in the most commonly used epoxy matrix. Although many examples exist in the literature [18,56], example performances of the jute thermoset composite including the authors’ previous work are discussed.

#### 3.4.1. Jute Fiber-Thermoset Composite

The unidirectional fibers can be incorporated into the thermoset matrix materials. For such types of composite fabrication, hand lay-up is the simplest and easiest way. Biswas, et al. [57] worked on developing unidirectional jute epoxy composite and mechanical performance is listed in Table 4. The composites displayed higher tensile strength and flexure strength compared to the epoxy matrix. Moreover, fiber distribution was not found uniform in the composites as shown in Figure 15.

Rahman et al. [58] studied the effect of fiber orientation and jute volume fraction in jute polyester composites on mechanical and thermal properties with a maximum volume fraction of 30%. With a suitable combination of fiber orientation and fiber volume fraction improved mechanical properties were obtained and this type of composite can replace wood-based structures and potential to be used as a thermal insulation material.

Liquid thermoset resins can easily penetrate through the fibers. However, when pressure is applied for uniform penetration, the fiber orientation might change affecting the load-bearing capacity of the composite. In order to overcome this limitation, jute fabrics are incorporated into the thermoset matrix for obtaining uniform properties in different directions.

#### 3.4.2. Jute Fabric-Thermoset Composite

The fibers are highly interlocked with each other in the woven fabrics compared to individual fiber or bundle fiber and their corresponding composites show higher strength. Therefore, when the number of jute fabric layers is increased within the matrix with good bonding, the corresponding composite shows increased performance. Different layers of jute fabrics [59] were incorporated into polyester through the hand lay-up technique and the findings are depicted in Figure 16a–c. It was clearly observed that tensile strength and modulus increased with an increase in the number of layers in the composite. However, with a higher number of layers, water uptake increased. A balanced approach should be adopted to get good mechanical properties without absorbing excessive water, which might affect the long-term mechanical performance.

Due to the presence of hydroxyl and other polar groups in various constituents of the natural fiber, the moisture uptake of the jute fabric-reinforced composite was higher compared to the synthetic polyester. This could lead to poor wettability of the fiber with the matrix, weak interfacial bonding and increased void content. Therefore, water repellant treatment of the jute fabrics or adding other fillers might reduce the water uptake. Jute based polyester composites targeted for a rooftop and fence for poultry were fabricated with different layers of jute-cotton blended fabrics (Jute/cotton: 50/50, 60/40, 70/30 and 100/0) and additives such as cashew nut shell (CNSL), nano-cellulose/nano-clay, and MEKPO hardener using a simple hand lay-up technique and their performance were analyzed [60]. With the increase of layer numbers, thermal conductivity increased except for 50/50 jute-cotton Figure 16d,e. A similar trend was also found in the case of water uptake behavior. However, the treatment with the nanoparticles brought a positive impact by reducing the thermal conductivity and water absorption. In order to further improve the jute polyester composite performance, the jute fiber was treated with alkali and nano-clay [3] and the results are shown in Figure 16f. A nano incorporation of 1% was found to be the optimum value for obtaining the maximum strength.

Jute fabric treatment by 10 to 15% NaOH was recommended to obtain the best mechanical properties in jute fabric-based epoxy composites [61]. Nath, et al. [62] incorporated the cenosphere from fly ash in woven jute-epoxy bio-composite to convert industrial waste into a value-added composite material. Addition of 5wt. % filler increased the tensile, flexural, interlaminar shear strength (ILSS) and hardness of the composite whereas the water absorption and thermal conductivity values showed an opposite trend. Nonwoven thermoset composites exhibit superior sound and heat insulation characteristics [63] as the fibers contain hollow lumen in their structure.

### 3.5. Jute Thermoplastics Composite

#### 3.5.1. Jute Fiber-Thermoplastic Composite

Jute is a cellulosic material and has free OH-, which leads them to absorb moisture. As a result, they can be easily affected by bacteria and fungus and rot easily. Furthermore, the cellulosic fiber easily burns. Therefore, jute fibers were processed with water, fire and water retardant treatments and composites were formed with Maleic anhydride–grafted polypropylene (MAgPP) using a hot press machine. In all cases, with the increase of fiber content (wt.%) in the composites, the mechanical properties improved as shown in Figure 17. However, the treatments caused degradation of the tensile strength properties compared to the untreated jute composite. Fiber breakage and pull-out were evident in the tensile fracture surfaces. Uniform fiber distribution within the matrix could be a major contributing factor in terms of obtaining better composite properties.

Short jute fiber (~5 mm) was chemically treated with alkali, potassium permanganate, and silane and mixed with PP to develop composites that can resist moisture absorption. All composites with the treated fibers showed better mechanical properties than that with the untreated fiber. Silane-treated jute-PP composite showed the lowest water absorption characteristics but the potassium permanganate-treated one showed the best mechanical properties. Depending on the application requirements, a suitable choice can be made [64]. Compared to other treatments, MAgPP and silane-treated jute–PP composites exhibited lower creep strains at longer durations [65].

#### 3.5.2. Jute Fabrics Thermoplastic Composite

There are varieties of woven fabrics (e.g., plain, satin and twill) where the fibers are highly interlinked are used to fabricate thermoplastic composites in a layered structure. Figure 18a shows the cross-section of a six-layer plain-weave jute fabric composite where the layers were fully embedded within the matrix with the layered structure [41]. Tensile strength and flexural strength increased with the number of layers (Figure 18b,c). During tensile failure, some fiber surfaces within the yarn showed not much evidence of matrix adhesion indicating a lack of fiber wetting by the matrix material. In some cases, evidence of poor adhesion was also noticed by a clear gap between the matrix and fiber. Other layered composites using different matrix materials such as polypropylene and nylon were also developed [41]. In the case of woven thermoplastic composite, the mechanical properties depend on a number of factors such as fabric weight, fabric structure, yarn count, types of fiber, processing technique, as well as the type of matrix material used.

Nonwoven jute fabric was also combined with thermoplastics as a promising reinforcement due to its flexible, lightweight and low-cost characteristics when compared to conventional woven fabrics. Sayeed et al. developed jute/polypropylene nonwoven reinforced composites using compression molding technique following film stacking strategy [49,66]. With alkali treatment of fiber, the composite showed improved mechanical properties. Tensile and flexural moduli could be maximized by tailoring the structure with the preferential and non-preferential alignment of the fibers with jute content between 23 and 33 wt.%. However, the layering sequence showed an insignificant effect on the tensile strength and elongation before breaking. Similarly, the rise in jute content enhanced the storage and loss moduli but the damping parameter showed a declining trend. Again, the layering sequence appeared as an influential factor in reaching the peak storage modulus [67].

### 3.6. Jute Based Polymer Biocomposite

Similar to other natural fibers, the major problem with the jute-based composite is that it cannot be easily recycled or biodegraded when compounded with the traditional non-biodegradable polymer matrixes. This only reduces the consumption of synthetic polymeric materials. However, in order to develop an eco-friendly green composite, both the fibers and the matrix materials must be biodegradable. Biodegradable composites with PLA films and jute nonwovens were developed by employing a film-stacking design in compression molding. Excellent tensile and flexural strengths were achieved by the composites but compared to the pure PLA, the composites showed slightly poorer thermal stability [68]. Meanwhile, completely degradable bio-composites have been made by Hu et al. [69] using short jute fiber for the automotive industry by increasing the volume fraction up to 70%. A higher volume fraction of jute fiber can effectively reduce the cost of the composite significantly. Short jute fibers and short PLA fibers were mixed with a jute volume fraction ranging between 60% and 70% and felt was made via carding and needle punching. The hot pressed molding process then used the felt as the starting material to fabricate the jute/PLA composite. The mechanical strength of the composites increased with the fiber volume fraction and good formability and processability were realized in manufacturing life-size truck liners for automotive applications. Other biobased jute composites using PLA have been recently reviewed [20]. Though the bio-based polymer composites are biodegradable, the strength is limited for a particular fiber, process and matrix, which can be tailored by hybridization and sandwich structure.

### 3.7. Hybridization with Jute Fiber

#### 3.7.1. Fiber/Filler/Matrix Combinational Structure

It is well known that the mechanical properties of the jute fiber-based composites are still inferior to synthetic fiber-based composites such as Glass Fiber Reinforced Composite (GFRP) or Carbon Fiber Reinforced Composite (CFRP) [70]. Jute fibers are mixed with other natural or synthetic fibers or fillers to obtain improved properties by exploiting the strength of multiple fibers/fillers. In some cases, the hybrid composites can add multifunctional characteristics suitable for certain applications. Hybridization can occur in the case of fiber as well as matrix material such as different polymer matrixes in the forms of liquid, granule, plate or sheet. Figure 19 presents some conceptual designs for hybrid composites with different combinations of fibers, fillers and matrixes or their arrangements with some cost estimation. Furthermore, fiber arrangements can be oriented at different angles (e.g., 0°, 45°, and 90°) or the fiber itself can be changed to different forms (e.g., particle, short fiber, long fiber, whisker, webbed fiber woven or nonwoven).

There are number of studies available in the literature on the hybrid composites with the addition of another natural fiber such as coir and wool in different types of matrix materials in combination with the jute fiber [71,72,73,74,75,76]. In general, the fibers are chopped into short fibers and mixed with a predefined ratio before adding to the matrix material. Figure 20 presents the hybridization effect on jute thermoplastic composites with different combinations of fibers (Jute, J; Coir, C and Wool, W) and matrixes (Polyethylene, PE; Polypropylene, PP). PE-based composites showed lower strength possibly due to the weaker matrix. On the other hand, the effect of wool fiber was more influential than the coir fiber when the matrix and fiber were fixed to PP and jute. The overall fiber content and the ratio between the fibers could affect the strength of the resulting composites. From the fracture surface of the composites, different fibers can be clearly identifiable and fiber breakage, fiber pull out, and weak bonding was also observed. Favorable entanglement between the fibers and better interaction with the matrix could be considered as the reason for improved strength in the hybrid composites. With a combination of multiple compatible natural fibers and a biobased matrix, a sustainable composite with tailored properties can be developed.

Synthetic glass/carbon fiber reinforced polymer composites are mostly used due to their uniform properties, high strength, durability but their non-biodegradability and non-renewability forces efforts to find alternatives. Natural fibers like jute can be incorporated with the GFRP/CFRP to minimize the use of synthetic fibers while maintaining reasonably good mechanical strength [70,77,78,79,80]. The content of synthetic fiber is more influential in defining the mechanical performance than the stacking sequence between the synthetic and jute fibers [81]. Therefore, appropriate tailoring of jute fiber composites with synthetic fibers could provide a balance between cost, performance and environmental impact [82].

Some nanofiller (carbon nanotube, graphene,) incorporation in the jute composite can increase performance which will create a new area of research [83,84]. Hybrid jute composites with a mixture of two matrixes such as reinforced polyvinyl chloride (PVC) and polypropylene (PP) were also prepared by compression molding. A mixture of 60% PP and 40% PVC matrixes was recommended for the best performance [85].

#### 3.7.2. Sandwich Structure

Jute fiber can also be used in the hybrid sandwich structure where the core part is thicker and lightweight and outer skins/panels are thin but stronger to provide necessary stiffness. A number of designs could be possible with nonwoven jute being the core or jute composite as the skin or even jute composite with hollow structure (lattice/corrugated) as the core as shown in Figure 21. These types of designs can provide lightweight structures with environmentally friendly jute material. Endless other designs are possible to use jute in sandwich composite simply expanding the imagination of the designers.

Aly-Hassan et al. [86] suggested an innovative composite design to be used as smart inclined roofs for houses in the snowiest countries. The sandwich structure was comprised of a nonwoven jute core with glass fiber reinforced composite (GFRP) skins. The key benefits of this design were that it is strong but lightweight, thermal insulator and noise suppressor. Furthermore, the structure was equipped with a heating element and a snow-repellent nanocoating system in the skin to break the contact layer between the accumulated snow and the roof for quick and automatic removal of the snow. Another sandwich design based on jute fiber was proposed for energy absorption in body armor application [87]. Jute-epoxy (JE) and Jute-epoxy with rubber core (JRE) composites were developed, and their energy absorption capacities were evaluated. The results revealed that the JRE composite absorbed 71% higher energy than the JE. The experimental damage patterns matched well with Finite Element Analysis (FEA).

In another design, epoxy-jute composite plates were arranged in a grid structure to be considered as the core where spruce wood acted as the skin [88]. Failure tests showed a number of modes including shear, panel fracture, separation of the core from the panel. Fiber pull-out and splitting were attributed to the underlying mechanism for the shear failure.

Similarly, a corrugated lattice sandwich structure made of epoxy-jute fiber was used as the core in another design [89,90]. Bending tests identified the shear buckling failure mainly in sandwich struts. Another example of the sandwich structure was given in [91] where Jute/PP composite plates reinforced with jute/PP commingled nonwoven fabrics were used as the facing materials and balsa wood, PET foam and PP honeycomb as the core materials with balsa wood being the best core under bending loading condition. This jute-based sandwich could be considered an eco-friendly, low-cost and lightweight structure for various engineering applications.

## 4. Emerging Applications of Jute Fiber

For the past many years, scientific research is mainly focused on developing jute fiber reinforced polymer composites and characterizing their physical, mechanical, and structural properties for various applications in a number of sectors such as textile products, automotive, construction, defense, packaging, home furnishing, fashion, transportations, and sports [18,21,92]. As jute fiber absorbs CO_2_, emits O_2_ and consumes relatively less energy during product development, hence from the sustainability viewpoint, jute-based products can be considered better than synthetic products. However, more recently jute fiber is being researched for diversifying the application areas. The following examples demonstrate the potential of jute fiber for future product development.

### 4.1. Nanomaterials from Jute

Jute fiber has now been converted into nanoparticles using chemical treatment and mechanical processing. CMC, MCC, CNC, and charcoal can be converted into nanoparticles to be used in different sectors such as heavy metal absorption, food packaging, drug delivery, etc. (Figure 22) either as an individual particle or by incorporating it into polymer matrix [93,94,95,96,97,98,99,100,101,102,103,104,105].

Nanocellulose has some distinct characteristics including high surface area, promising rheological properties, ability to absorb water, crystalline structure, favorable surface chemistry, barrier properties and nontoxicity, which make it suitable to be used in packaging film and food applications [103]. Jute itself, as well as its constituents, have absorbing capacity and therefore absorbing materials can be developed for efficiently separating heavy metals using jute-based nanomaterials [98]. Furthermore, when nanocellulose is incorporated into the matrix material, the nanocomposite film showed superior strength and physical properties [97,106]. An appropriate level of cellulose nanofibril content can improve the interaction with the matrix leading to better mechanical properties. Possible other prospects of jute fiber in a host of other nanotechnological applications are discussed in recent literature [102].

### 4.2. 3D Printing with Jute

3D printing has found applications in producing products with complex shapes eliminating the need for assembling, though limited by the choice of materials and speed of production [107,108,109,110]. Initially, 3D printing was limited to only a handful of polymeric materials but thanks to the scientific research in making materials such as metals, ceramics and composites in different forms including powder or filament. Conventional composite fabrication is difficult for complex shapes; therefore, 3D printing opens up new opportunities. Jute fiber can be used to develop composite filament for printing jute-based products. Figure 23 shows process steps in manufacturing 3D printed biobased jute composite.

Matsuzaki et al. [111] invented a unique method of 3D printing of continuous jute fiber-based PLA thermoplastic composite using fused-deposition modeling where both the polymer filament and the twisted jute yarns were individually fed to the printer. The fibers were unfirmly embedded within the matrix as a reinforcing material. Carbon fiber-based PLA composite was also produced following the same procedure for comparison. Although jute fiber-reinforced composite showed slightly better tensile strength than the pure polymer but was inferior to the carbon fiber-based composite. The volume fraction of the jute fiber was quite low (6.1%). However, some success was achieved when continuous flax fiber/PLA (cFF/PLA) composite filaments were made by coating the fiber with PLA using an extrusion process and printed by the composite filament in a simple 3D printer [112]. Unform distribution of the yarn with the matrix was observed in the printed samples. Significant improvement in tensile strength and modulus was achieved compared to the published results on 3D printed continuous natural fiber printed composites.

Perez et al. [113] compared the strength and fracture behavior of FDM 3D printed ABS with different fillers including 5 wt.% jute fiber. Although the jute fibers were embedded within the ABS matrix but the strength of the jute-ABS composite was slightly lower than the pure ABS. Evidence of higher porosity and poor bonding might be the cause of the poor result.

Other than composite filament or individual fiber-polymer filament feeding to the printer, an alternative approach was explored by Franco-Urquiza et al. [114]. PLA-fused filament was deposited onto jute fabrics. The jute fabric was modified with flame retardant additives, and adhesives to gain improvement in mechanical properties. However, the composite did not show any improvement over the pure PLA in terms of stiffness or strength even though fabrics were well organized and bonded with the matrix. A further detailed investigation was suggested by the authors to obtain a better composite.

Nano fiber/cellulose or activated carbon collected from the jute fiber can be added to the 3D printer filament for applications in aerospace, packaging, and medical appliance such as scaffolding [115]. Jute fiber treated with an antibacterial agent can be encapsulated in polymers for providing a slow-release antibacterial effect over a long period of time. Therefore, jute fiber-based 3D printed products can lead to completely new research directions.

### 4.3. Electronics, Energy Storage and Sensing

Jute fiber containing cellulose, hemicelluloses, lignin and a small amount of ash makes it a suitable raw material for activated or non-activated carbon preparation utilizing a number of methods such as pyrolysis with or without chemical and physical activations. Having a fine porous structure and a large specific surface area, carbon-derived from jute could be used in a number of applications such as energy storage, water treatment, and sensing [116,117,118,119] due to its capacitance, filtering and electronic properties respectively.

Activated carbon was synthesized from jute fibers with KOH activation at a high temperature of 700 °C for hydrogen storage application [120]. The activated carbon retained channel-like fibrous structures with increased surface area and micropore volume showed promises of an increased hydrogen storage capacity. Another study investigated the potential of micro-mesoporous carbon material derived from jute fiber as an anode in lithium-ion batteries [121]. Manjakkal et al. [122] designed and developed an energy-autonomous system containing a jute fiber-based supercapacitor (SC) and temperature and humidity sensors. The SC was charged using a solar cell while the biobased sensor was powered using the SC. This system showed huge potential as an eco-friendly solution in applications like wearables and smart packaging for food quality monitoring. Biodegradable conducting polymer composites were developed by adding jute fiber to Polyvinyl Alcohol (PVA)/multi-layer graphene (MLG)/multi-wall carbon nanotube (MWCNT) [123]. The composites retained viscoelastic nature as found in a dynamic mechanical analysis. Furthermore, the composite showed high electrical conductivity and effectiveness in shielding electromagnetic interference.

Success was made in developing Transparent jute fiber (TJF) from delignified jute fiber as a replacement for transparent wood as the TJF has the ability to match the properties of the transparent wood [124]. Owing to the processing simplicity and lower processing cost along with good optical and mechanical properties, the TJF might find applications in smart devices, for example, light-emitting diodes (LEDs) and solar cells [125]. Multiple recyclable transparent film was also produced from waste jute fabrics using the ionic liquid (IL) assisted regeneration process [126]. This film has the potential to be used in packaging materials or in electronic devices for information storage. Therefore, these types of jute base advanced material will lead to a novel area of research in the near future.

### 4.4. Biodegradable Packaging

#### 4.4.1. Sonali Bag—Solution to Single-Use Plastic

Single-use synthetic plastics for instance plastic carrier bags have become an integrated part of daily life over the past many years. However, there is a growing concern worldwide that their uncontrolled release into environment can cause significant damage to marine life, plastic pollution on the earth and imbalance in the ecosystem. In many countries in the world particularly in Europe, single-use plastic will be banned in the future, and it will be replaced by more environmentally friendly biobased and biodegradable materials [127]. More recently, a team in Bangladesh developed an alternative low-cost biodegradable solution to polyethene bags named the “Sonali Bag” from modified jute cellulose, which can be biodegraded in soil within three months and showed similar or better mechanical and physical properties compared to the synthetic polyethene bags [128]. This new biobased packaging can revolutionize the packaging industry worldwide and solve the increasing problem of plastic pollution (Figure 24). For this type of product, mass manufacturing at a low cost would bring commercial success.

#### 4.4.2. Bioactive Jute for Food Packaging

Due to increasing demand for sustainable food packaging materials, jute fiber was treated with Red Grape Pomace Extract (RGPE) as an activation material [129]. The RGPE impregnated jute fiber showed antioxidant and antimicrobial properties against a number of microorganisms such as *Pseudomonas aeruginosa*, *Escherichia coli*, *Staphylococcus aureus* indicating its potential as a food packaging material with better preservation capacity. These types of natural reagent-based research need to be flourished for sustainable product development.

### 4.5. Cotton/Wool Replacement

#### 4.5.1. Absorbent Cotton for Medical Disposables and Sanitary Products

Owing to the increasing demand for hygiene and wound dressing products, jute can be considered a cheaper replacement option for cotton fiber in disposable absorption products (Figure 25). Through chemical treatment, lower grade jute can be transformed into absorbent cotton which can be further modified to be used as wound healing bandages, sanitary napkins or baby napkins. To develop these types of products, the jute fiber needs to be treated with anti-bacterial and anti-fungal agents. Based on this contemplation, AgNO_3_ and Co can be used for increasing their anti- bacterial activity. Therefore, such types of feasibility studies have already been undertaken by certain researchers [130,131]. Sharma et al. [132] developed disposable pads using waste jute blended with Superabsorbent polymer (SAP) in order to increase water absorbency. Various functional tests demonstrated the potential of jute blended with SAP as sanitary pads at an affordable cost. In a recent survey [133], it was found that women are inclined to adopt jute-based sanitary pads. Further studies are required to commercialize jute-based absorbent product concepts. Therefore, this is an open research topic for the advanced development of medical appliances.

#### 4.5.2. Jute-Cotton Blending

Recent research shows that blending jute with cotton could produce sustainable textile products. The blended textile has already found applications in durable products like denim and polo knitwear [134]. In general, chemical modification and bleaching are applied to the jute fiber to make it softer. Research is being carried out on jute-cotton blended textiles to bring certain functional properties by applying different surface treatments for instance atmospheric plasma treatment for improving wicking and dyeability properties whereas chitosan treatment for incorporating antibacterial property [135,136]. Natural fibers like jute can produce less ecological, water and CO_2_ footprints [137], which will have an overall positive environmental impact. Other benefits of jute-cotton blending include reducing dependency on cotton, producing textiles at a lower cost, diversifying the use of jute, contributing to the local economy of the jute-producing countries [138] and helping in strengthening the already established garment sector in lower-middle-income countries like Bangladesh. The people in western countries are gradually becoming inclined toward using garments made of more sustainable materials. Therefore, a significant growth in jute-cotton blended products is expected in the near future.

#### 4.5.3. Woolenization

The idea of jute fiber woolenization has been explored by converting the jute fiber into wool-like material simply using alkali treatment. It was found that with 15% NaOH treatment, a significant modification in the jute structure was possible to make such transformation [139]. The effectiveness of this technique was demonstrated by developing commercial prototypes like handbags. Further efforts are needed to identify the market opportunities and to diversify the use of jute in the fashion industry.

### 4.6. Jute Engineered with Nanomaterials

The lower strength of jute, sensitiveness to moisture and poor interaction with polymer matrix material in composite compared to other synthetic fibers are still some of the weaknesses that hinder the progress towards mass use of jute fiber. Among different strategies, many researchers suggested the functionalization of jute fibers with nanoparticles (NPs) to overcome the challenges. Some examples of NP functionalization of jute fibers are found in the literature.

#### 4.6.1. Graphene

In recent years, a number of studies reported on the functionalization of jute fiber with graphene or graphene-like materials particularly by a research group at the University of Manchester [140,141,142,143,144]. The modified jute fiber and epoxy composite showed improved mechanical properties than other natural composites. Stiffness and tensile strength were improved by ~324% and ~110%, respectively, when compared to the untreated jute fiber composites. This improvement was assigned to various reasons such as individualization of jute fibers, improved fiber packing and new fiber structure within the composites and strong adhesion of the fibers with the matrix. Graphene oxide nanoplatelets (GONPs)-grafted jute fiber was used to develop thermoplastic (polypropylene) based composite [145]. The results showed significant improvement in mechanical and thermal properties of the composites, which was attributed to effective interlocking between the surface treated jute fiber and the matrix. This type of composite might attract commercial interest in several fields such as automotive, aerospace, architecture, and household products and possesses the potential to substitute the conventional synthetic fiber-based composites such as GFRP or CFRP. Following a similar process of jute fiber functionalization by graphene, the development of composites with other matrix materials can be explored. The nanoengineered jute fiber might find applications, particularly, in electronic devices. Therefore, this will open up a new era of jute research and might attract a renewed interest among the researchers.

#### 4.6.2. Carbon Nanotubes (CNT)

Carbon nanotubes can be deposited on the jute fibers to make them conductive by using the dip-drying method [146] with the assistance of oxygen plasma. This technique produces uniform and efficient coating strongly attached to the surface of the jute fibers. The coating enhances the crystallinity and mechanical strength of the modified fibers. The functionalization of the jute fiber further improved electrical conductivity, thermal stability and flame retardancy. Therefore, jute fiber functionalized with CNT could open up opportunities to use in smart electrical and electronic devices.

#### 4.6.3. SiO_2_ Nanoparticles

Jute fiber and polyamide 11 (PA11)-based novel biodegradable composites were developed by modifying the jute fiber with SiO_2_ nanoparticles of approximately 100 nm [147]. Clear evidence of the nanoparticles attached to the fiber was found. The modification of jute fibers enhanced their tensile strength, Young’s modulus and impact strength by 58.3%, 45.9% and 11.7%, respectively, indicating better stiffness and toughness than the unmodified jute fiber-reinforced composite. Furthermore, significant improvements in thermal stability and water absorption characteristics were also found due to strong interaction and adhesion between the fibers and the matrix surface. Therefore, engineered jute fiber with SiO_2_ nanoparticles showed great potential as a biobased composite in real-life applications.

Another study was performed by Araújo et al. [148], where jute was treated with CaO and SiO_2_ nanoparticles in order to embed multifunctional characteristics such as UV protection, hydrophobicity, antibacterial activity and wash durability. The combination of CaO and SiO_2_ instead of CaO only showed improvement in incorporating the multifunctional characteristics. The treated fabric might find potential applications in the area of personal protective equipment.

## 5. Challenges and Potential Future Directions

More recently, jute fiber, its derivatives and associated composites have enjoyed major successes in research and commercially in several application areas. A SWOT (Strength, Weakness, Opportunity, and Threat) analysis of the jute fiber as a technical raw material is presented in Figure 26. By no means this is a complete list of all strengths, weaknesses, opportunities and threats. It is expected that in the future, the items in strength and opportunity parts will grow whereas the items in weakness and threat parts will decrease. In this section, some of the important challenges of natural fibers [149], particularly, jute fibers and possible actions that can be considered are briefly discussed.

### 5.1. Inferior Properties of Jute Fiber

Despite having a number of favorable characteristics, the jute fiber is facing a number of challenges to enter into the global market as a preferred biobased sustainable fiber. However, some of the challenges such as lower strength than some natural fibers (e.g., flax), thermal degradation at low temperature, sensitiveness to water absorption and variations in fiber quality and performance needs to be addressed. Fiber modifications with a variety of innovative chemical or physical treatments such as engineering fibers with different nanoparticles would provide some solutions towards improving the functional characteristics of the jute fiber.

### 5.2. Challenges in Composite Processing

With the traditional composite manufacturing techniques, reliability on the quality of the jute-based composite products still remains a major issue. New composite processing techniques should be developed to obtain high-quality and consistent products with high fiber content with an emphasis on optimizing processing conditions. One option could be changing the traditional granule, sheet or plate forms of matrix to fiber, which can increase the fiber volume ratio within the composite. New technologies need to be developed for ensuring composite quality. While improving the quality of the products, the cost of processing or the product cost per unit product has to be kept minimum. Some attempts have been made to print jute composite by the 3D printing technique, but still, void formation and low fiber content problems must be resolved.

### 5.3. Issues with Composite Performance

Some of the inherent weaknesses of the jute fiber are also transferred to its associated composites. Jute composites still show poorer strength than the popular synthetic fiber and some natural fiber composites. The issues in the composite are attributed to poor bonding with the matrix. Furthermore, limited fiber volume (20% to 40%) in the composite hinders it from achieving lightweight but strong and greener composites. The idea of hybridization along with fiber surface treatment, which can help in improving the properties of the composites by promoting a strong bonding of the fiber can allow it to compete with other high-performing synthetic-fiber-reinforced composites. Nanoparticle treatment of the fiber can also help in facilitating jute-polymer bonding, which can accommodate high fiber volume within the composites. At the moment, a limited number of biobased and biodegradable polymer matrixes are available. Further research efforts are required in developing new biobased polymer matrixes. The chemicals used in fiber treatment might cause some environmental impacts. Therefore, research in finding feasible natural biobased extracts and natural dyes for fiber treatment needs to be explored. A combination of biobased and biodegradable polymer matrix and jute fiber treated with biobased chemicals or other environmentally friendly but less energy-intensive physical treatments can make a truly green composite.

Water uptake by the composite could cause swelling of jute fiber leads to fiber detachment from the matrix interface and eventually affects the performance. New technologies for waterproof coating could minimize this issue [59]. Studies are still lacking in assessing the mechanical performance of the composites after physical or thermal aging or long-term performances like creep behavior [65].

### 5.4. Lack of Product Diversification

The quantity of jute-related products and their share of the global market still remains small possibly due to inadequate international promotion and lack of product diversification. New ideas need to be generated with target applications for both high-value and mass-consumed products. Although several new ideas for jute-based products such as cotton replacement and nanocomposite development have attracted considerable research interest, the knowledge gained in research needs to be transferred to the industry for commercialization. Extensive marketing efforts are required to present the jute products to international customers and traders such as displaying in international exhibitions like Dubai Expo in 2021–2022. This will attract international entrepreneurs to investing in jute-based product development and create opportunities for new research projects with international collaborators. Countries like Bangladesh and India, which are the major jute exporters, should conduct collaborative research to develop novel ideas to support diversification of the jute products in the international markets. Combined policy development by the jute producing countries can contribute to its widespread use in both local and international markets. It is expected that the emergence of new technologies in fiber and composite processing will support the quality improvement and market development for the diversified jute products.

### 5.5. Recycling, Biodegradability and Circular Economy Issues

In spite of significant environmental and cost benefits, mismanagement of the end-of-life of natural fiber-reinforced composites raises a lot of concern for the industry possibly due to the fact that a large amount of plastics get incinerated or go into landfills, particularly, thermoset plastics, which cannot be recycled easily. Even it is recycled, the cost of recycling could be higher than the value of the material recovered. Therefore, innovative recycling processes or strategies need to be developed. Biopolymer-based jute composites are subjected to biodegradation rather than extending the composite life either by recycling or reusing or remanufacturing. This area needs special attention for further research. The circular economy approach advocates for transforming waste into resources in a cost-effective manner. The circular economy system should be applied in managing composite end-of-life through effective material recovery and applying new technologies for recycling wide varieties of jute composites. Recycling of industrial waste of jute and polypropylene fibers from the carpet industry showed some promising results [150]. In order to implement the circular economy paradigm for biobased composites, the life cycle framework has to be considered [151,152]. Assessing the environmental impact and life cycle cost at every stage of the product life cycle would help in making decisions for sustainable product development [153].

## 6. Conclusions and Outlook

In this paper, current progress in the research about jute fiber and its associated materials such as fiber-reinforced polymer composites has been reviewed and its future potential for product diversification has been critically analyzed. As a natural fiber, wide variations in microstructural and mechanical properties exist in the jute fiber, yarn or fabric. It is clear that other than traditional applications of jute as mainly a woven packaging material, it has found wide applications from macro to nanoscale particularly as a reinforcing fiber for polymer composite products due to the increasing demand for using sustainable materials.

Composite materials can be developed using combinations of both thermoset and thermoplastic materials with different forms of jute such particles, short fibers, long fibers and woven/nonwoven fabrics. Different types of chemical and physical treatments of the fiber have been developed to improve the compatibility between the fiber and matrix, which can improve the performance of the fiber or associated composites. Combining jute with a biobased polymer matrix made it possible to develop eco-friendly and green composites. Hybridization of jute-based composites by adding other natural fibers or synthetic fibers or particles shows further improvement in performance.

Overall, the jute fiber has the ability to replace environmentally harmful man-made fiber (e.g., glass fiber) and, in the near future, diversified high-value applications of the jute fiber and its derivative nanomaterials in biodegradable packaging, fashion, electronics, medical and energy sector are expected to increase significantly. Hopefully, jute will find its glory again by capturing new markets with the offerings of fashionable and eco-friendly products for the growing number of environment-conscious customers.

## Figures and Tables

**Figure 1 polymers-14-01445-f001:**
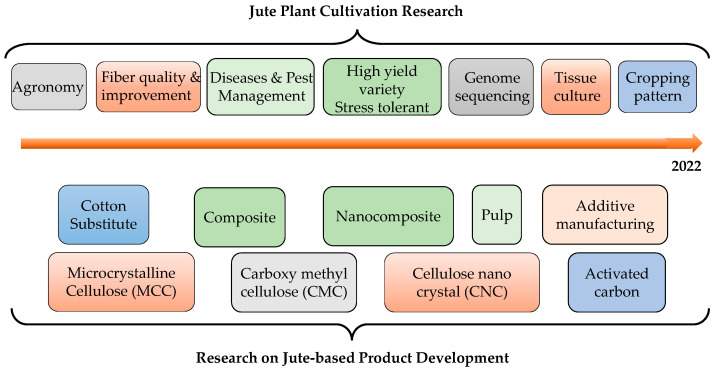
Trend of research on jute plant and fiber.

**Figure 2 polymers-14-01445-f002:**
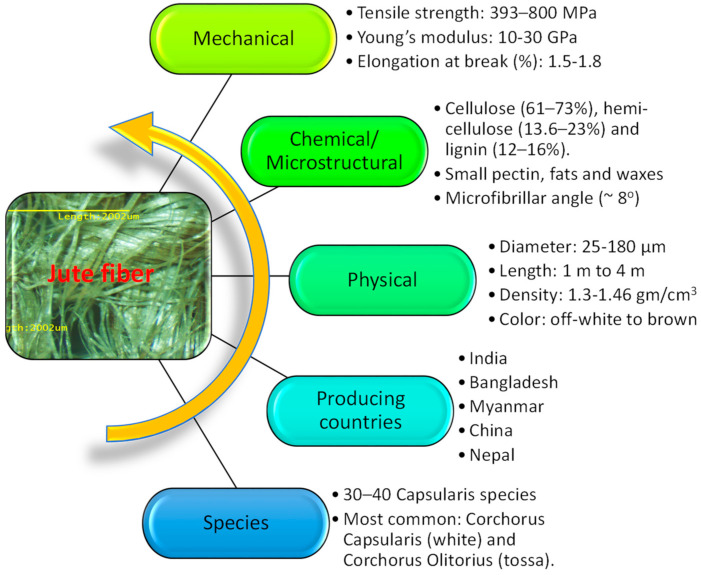
Facts and figures about jute. (Information taken from [6,7,8,9]).

**Figure 3 polymers-14-01445-f003:**
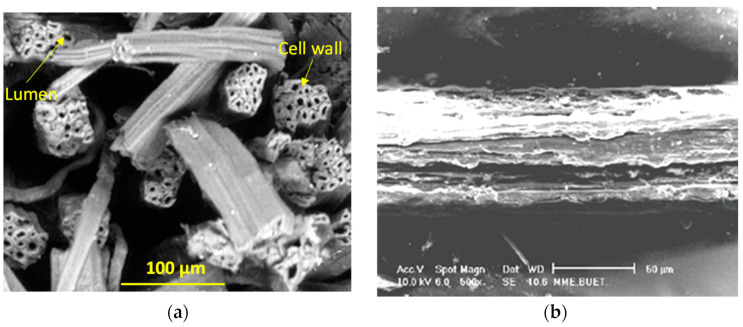
Jute fiber morphology: (**a**) cross-section and (**b**) surface [9].

**Figure 4 polymers-14-01445-f004:**
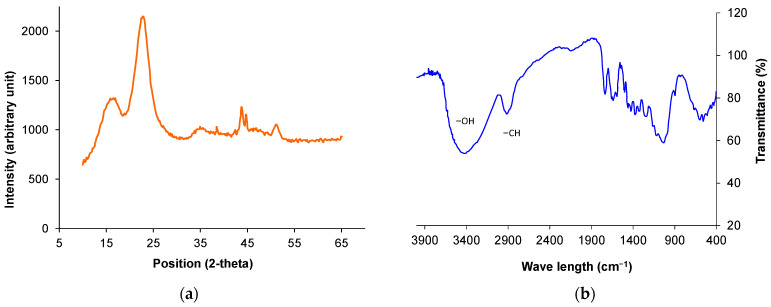
Microstructural characteristics of jute: (**a**) X-Ray diffraction and (**b**) FTIR spectra [28].

**Figure 5 polymers-14-01445-f005:**
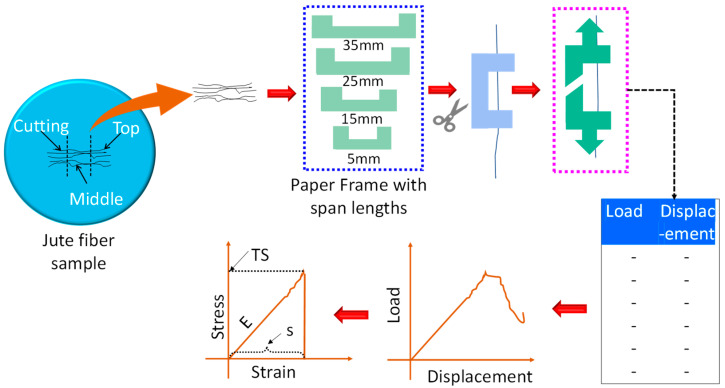
Single jute fiber characterization. TS: Tensile strength, E: Young’s modulus.

**Figure 6 polymers-14-01445-f006:**
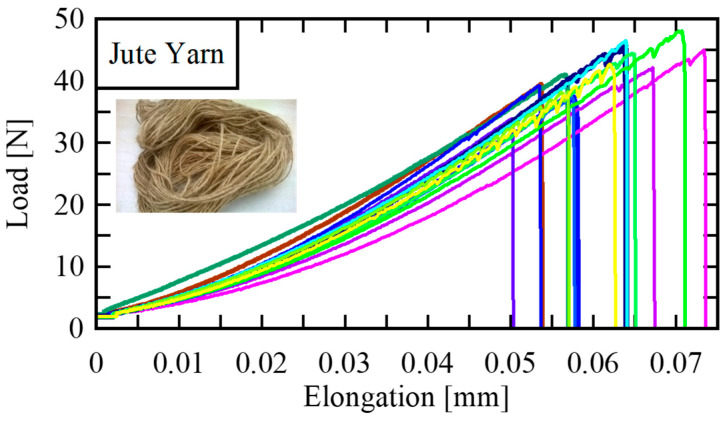
Typical load-elongation plots for jute yarn [2].

**Figure 7 polymers-14-01445-f007:**
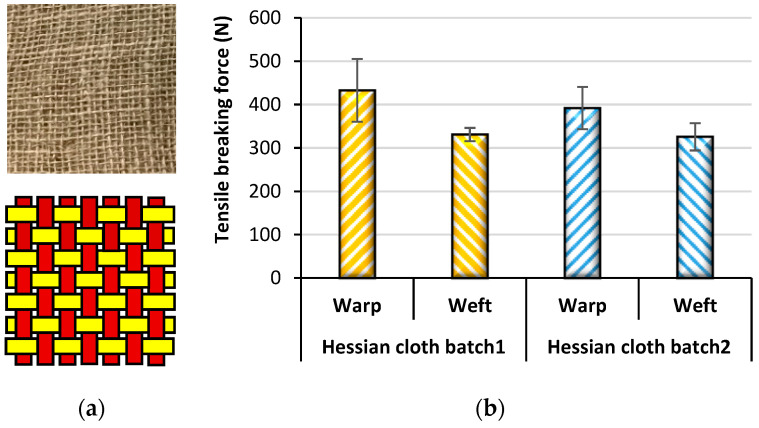
(**a**) Hessian Jute fabric with schematic representation of plain weave and (**b**) differences in tensile breaking forces of the jute fabrics in different directions from two different batches.

**Figure 8 polymers-14-01445-f008:**
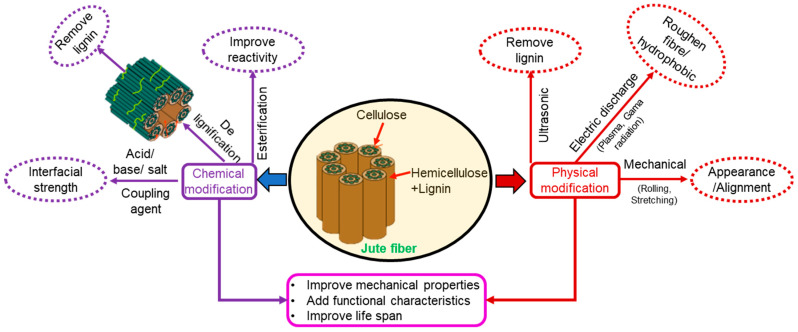
Types of treatments on jute fiber and their effects on fiber performance [4].

**Figure 9 polymers-14-01445-f009:**
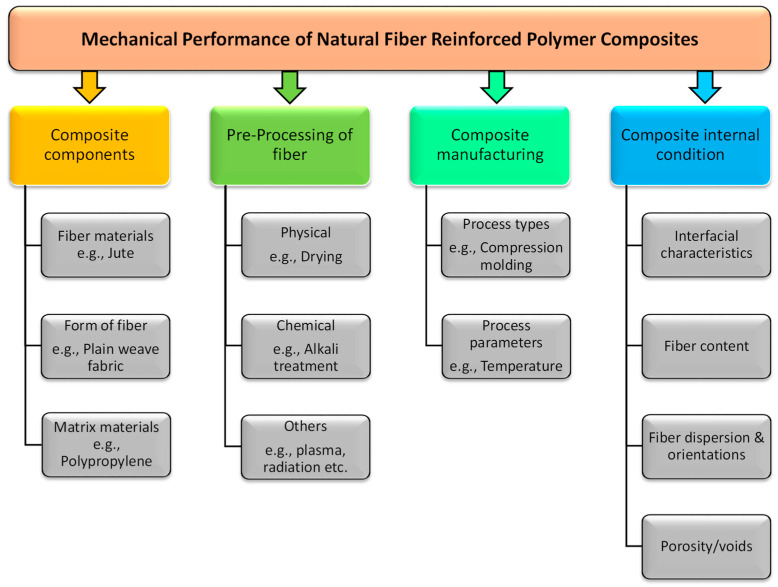
Factor influencing natural fiber reinforced composites [40].

**Figure 10 polymers-14-01445-f010:**
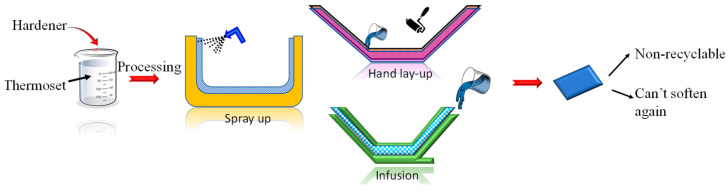
Thermoset matrix material in composite fabrication.

**Figure 11 polymers-14-01445-f011:**
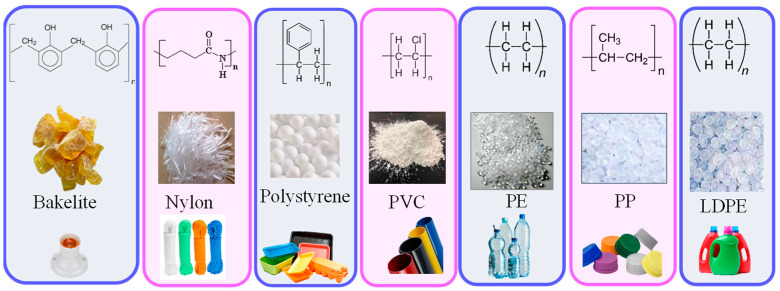
Different categories of thermoplastics and their popular applications.

**Figure 12 polymers-14-01445-f012:**
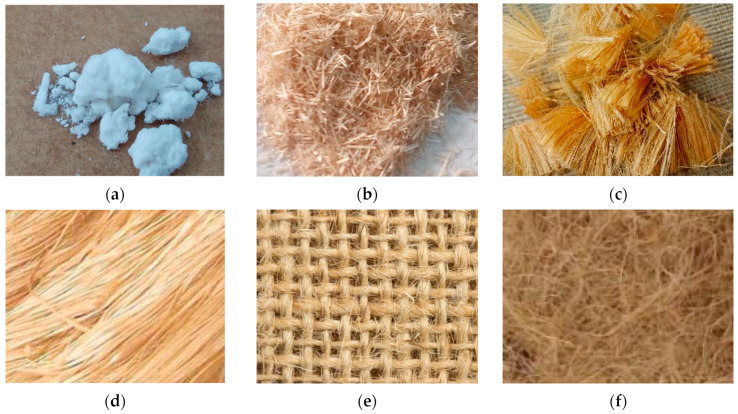
Different forms of jute fiber reinforcements used for fabricating polymer composite: (**a**) Particles, (**b**) Short (**c**) Semi-long (**d**) Long (**e**) Fabrics (**f**) Nonwoven.

**Figure 13 polymers-14-01445-f013:**
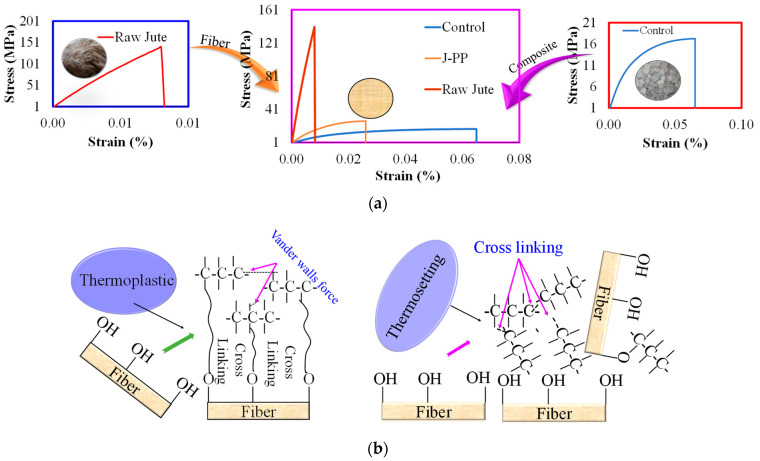
(**a**)Effect of the fiber in the matrix materials (**b**) cross-linking between the fiber and two types of matrix materials. Control: pure polypropylene; J-PP: Jute-polypropylene composite [54].

**Figure 14 polymers-14-01445-f014:**
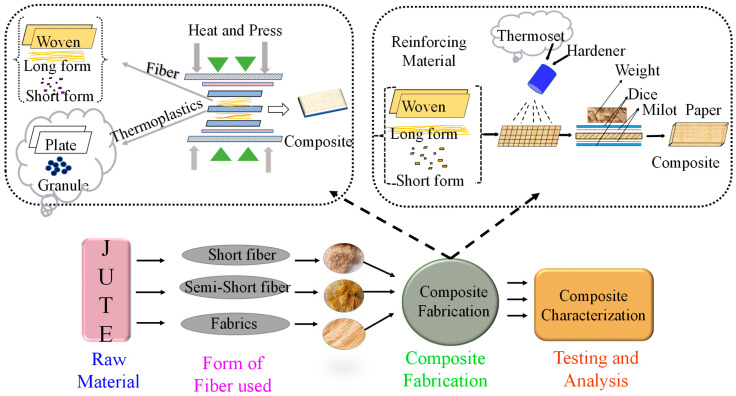
Composite fabrication using the hot press and hand lay-up techniques.

**Figure 15 polymers-14-01445-f015:**
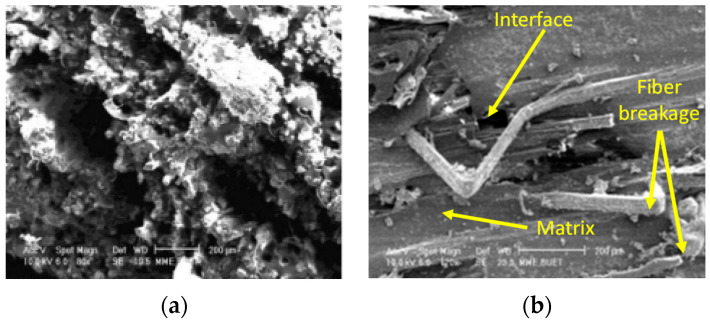
(**a**) Tensile and (**b**) flexural fracture surfaces of jute-epoxy composite [57].

**Figure 16 polymers-14-01445-f016:**
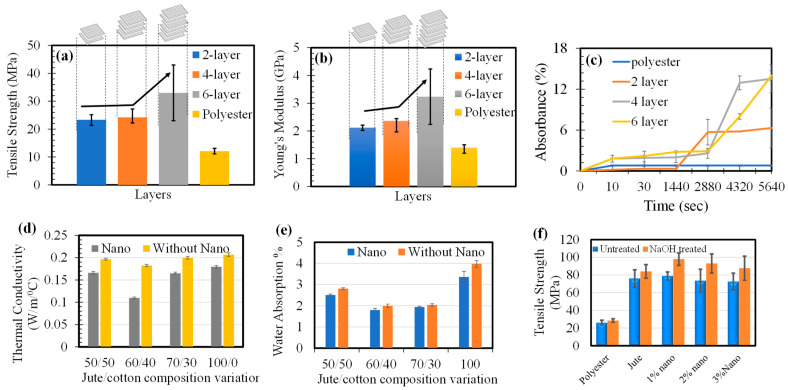
Performance of jute-thermoset composites (**a**) Tensile strength (**b**) Young’s modulus (**c**) water uptake of different layers jute polyester composite; (**d**) thermal conductivity and (**e**) water absorption of jute cotton polyester composites; (**f**) jute polyster composite with treated and untreated jute fiber [3,59,60].

**Figure 17 polymers-14-01445-f017:**
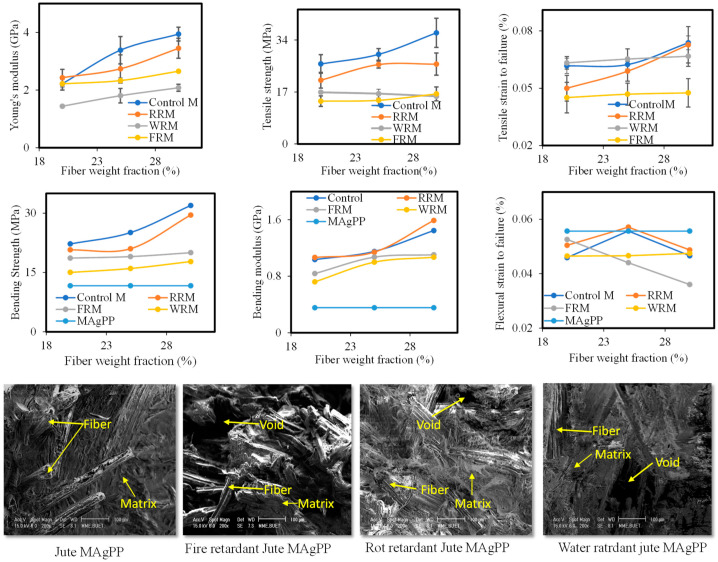
Mechanical properties and fractured surface characteristics of jute fiber reinforced thermoplastics composite fire-retardant (FRM), rot-retardant (RRM) and water-retardant (WRM) treatments on the jute fiber. Control M indicates no fiber treatment [43].

**Figure 18 polymers-14-01445-f018:**
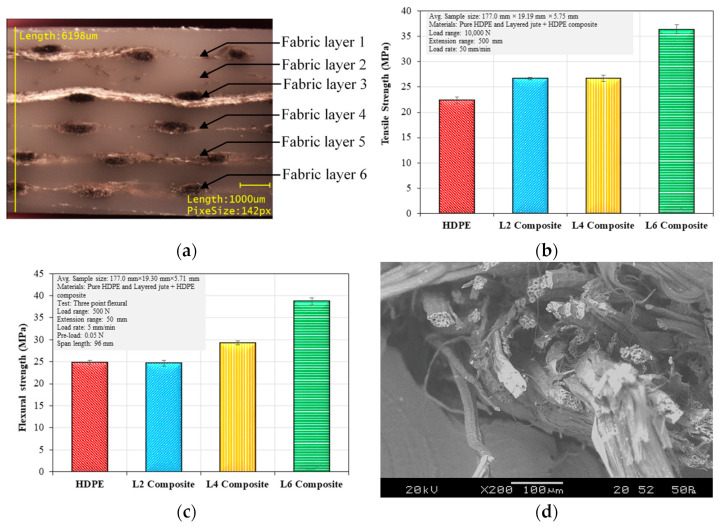
Mechanical properties jute-HDPE composites with varying number of jute layers: (**a**) layered composite cross-section (**b**) tensile strength (**c**) flexural strength and (**d**) tensile fracture surface [41].

**Figure 19 polymers-14-01445-f019:**
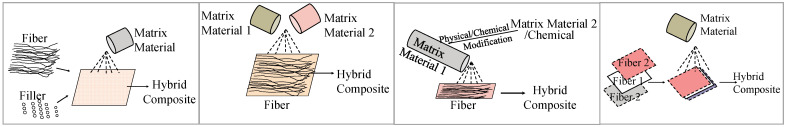
Schematic diagrams of hybrid composite designs and some cost estimation [54].

**Figure 20 polymers-14-01445-f020:**
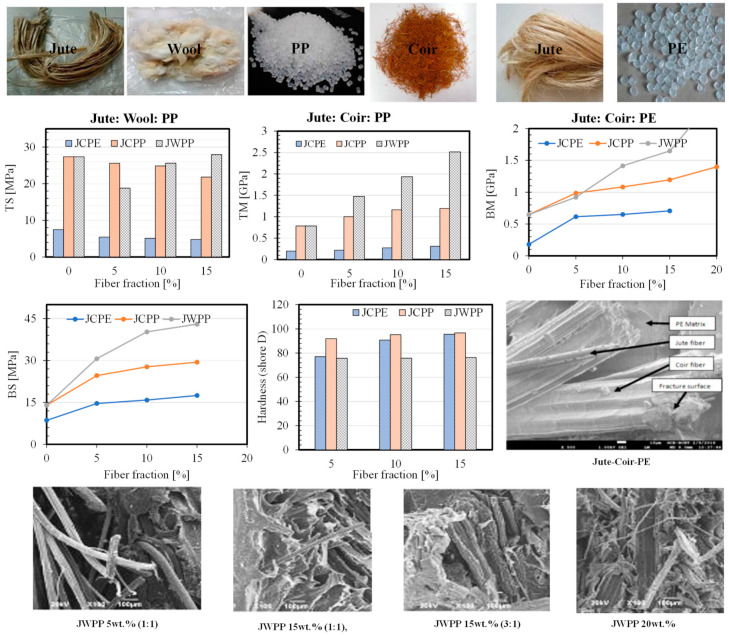
Effect of natural fiber hybridization on jute composite [71,72,73].

**Figure 21 polymers-14-01445-f021:**
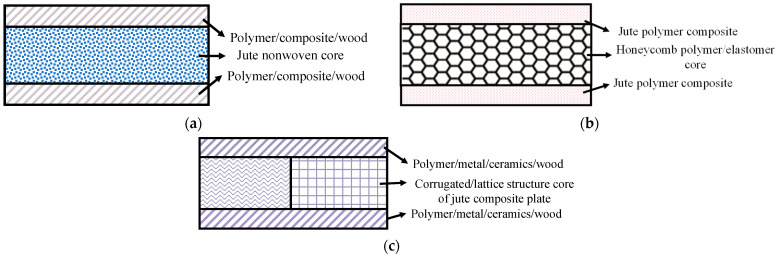
Some examples of sandwich type jute-based composite design (**a**) jute as core (**b**) jute composite as skin and (**c**) jute composite lattice/corrugated structure as the core.

**Figure 22 polymers-14-01445-f022:**
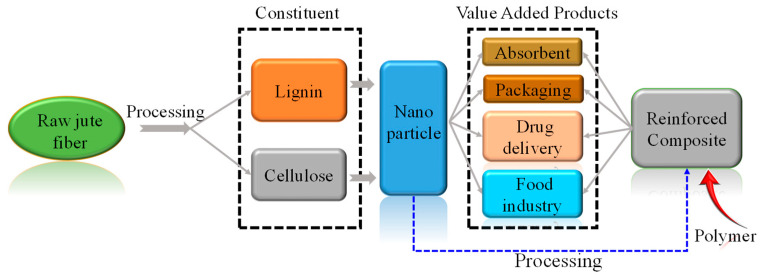
Examples of product development with jute nanoparticles.

**Figure 23 polymers-14-01445-f023:**
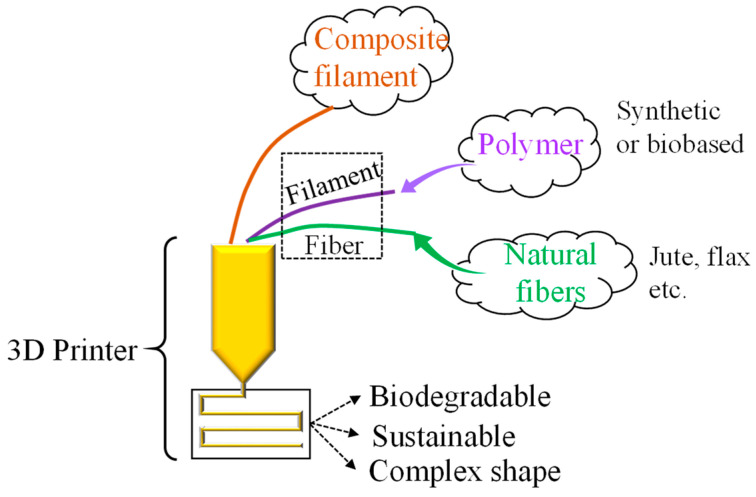
Natural fiber incorporation in 3D printing filament.

**Figure 24 polymers-14-01445-f024:**
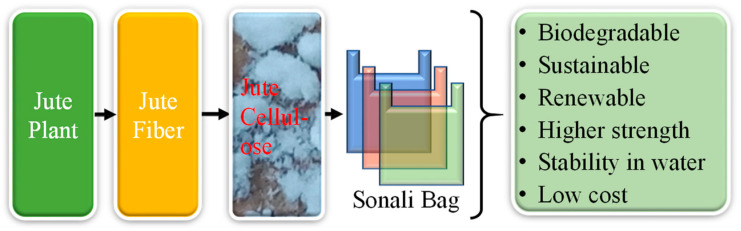
Jute-based biodegradable polybag production.

**Figure 25 polymers-14-01445-f025:**
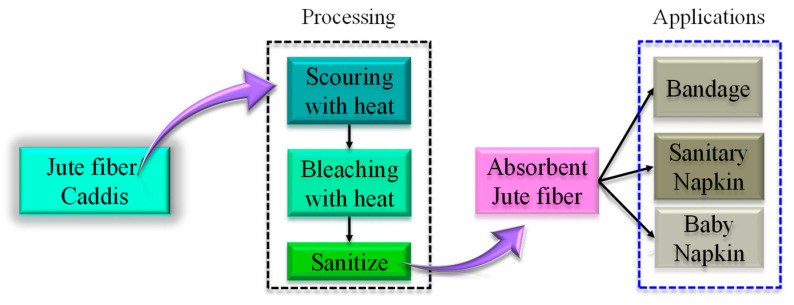
Jute fiber converted into an absorbent cotton-like product for medical accessories and personal hygiene products.

**Figure 26 polymers-14-01445-f026:**
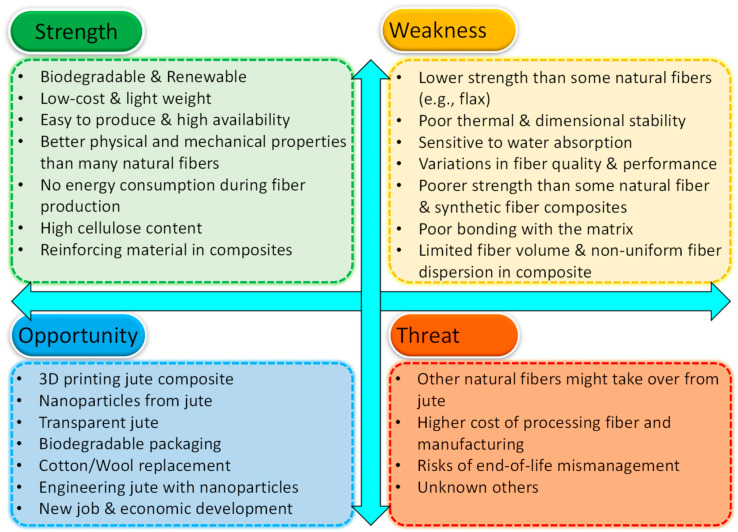
SWOT analysis of jute fiber and its associated products.

**Table 1 polymers-14-01445-t001:** Tensile properties of jute fiber/yarn/fabric.

Sample Name	Tensile Strength (MPa)	Tensile Modulus (GPa)	Strain to Failure (%)	Reference
Jute yarn	33–48	0.61–0.73	5–7	[2]
Jute Fiber	200–600	9–27	1.22–4.93	[29]
Jute Fiber	340–384	-	2.63–4.43	[30]
Jute Fiber	307–399	24.7	1.4–4.7	[31]
Jute Fiber	500–1000	30	1.5–2.5	[32]
Jute Fiber	249–314	35–48	0.6–0.9	[33]
Jute yarn	26–58	0.244–0.404	5.91–9.49	[34]
Jute Fabrics (Warp direction)	12–15	-	9–9.5	[35]
Jute Fabrics (Weft direction)	11–12	-	4.58–7.36	

**Table 2 polymers-14-01445-t002:** Specification of jute fabric.

Parameters	Value	Unit	Test Standards
Weave design	1/1 (plain)		BS EN 1049-2:1994
Warp	35	Ends per 100 mm	
Weft	31	Picks per 100 mm	
Weight	177	g/m^2^ (GSM)	BS 2471:2005

**Table 3 polymers-14-01445-t003:** Physio-mechanical properties of common thermoplastics and thermoset matrix materials.

Type of Matrix	Matrix Materials	Density (gm/cc^3^)	TS (MPa)	TM (GPa)	BS (MPa)	BM (GPa)	References
Thermoplastics	PP	0.89–0.92	32	0.9	35	1.0	[50,51]
	LDPE	0.89–0.92	29	0.20	0.09	0.1	
	HDPE	0.94–0.97	25	0.6–1.5	32.5		
	Nylon	1.14	50–90	1.3–4.2	20–150	1.3–3.7	
Thermoset	Epoxy	1.10–1.40	73	5	60	-	
	Polyester	0.90–1.27	20.40	3.3–3.9	45	3.3	
Biodegradable	PLA	0.90–1.27	25–66	2.3	48–110	0.3–3.6	
	Starch		4.48–8.14	0.16–0.29			

TS = Tensile strength, TM = Tensile Modulus, BS = Bending Strength, BM = Bending modulus.

**Table 4 polymers-14-01445-t004:** Tensile and mechanical properties of jute epoxy composite.

Matrix/Composites	TS (MPa)	TM (GPa)	Strain to Failure (%)	BS (MPa)	BM (GPa)
Epoxy	68.6	73	-	96–117.3	2.1–2.4
Jute-epoxy (52 wt.%)	216 ± 1.02	31 ± 1.34	0.78 ± 0.05	158 ± 18.90	18 ± 1.92

Note: TS: tensile strength; TM: Tensile modulus; BS: Bending strength and BM: Bending modulus.

## Data Availability

The data presented in this study are available within the article.

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
