# Peer review of "Current Development and Future Perspective on Natural Jute Fibers and Their Biocomposites"

_polymers, 2022, doi:10.3390/polym14071445_

Round 1

Reviewer 1 Report

The authors have prepared a review article entitled “Current Development on Natural Jute fiber and its Bio-Composites”. The article has some interesting results. A significant effort was made to prepare this article, which is interesting. This article would show a significant impact on the nanotechnology, materials science, and biomedical communities. However, there are some issues with the manuscript as it stands (detailed below) and these need to be addressed before the manuscript can be considered further. For instance, the authors should include more literature on other biomaterials to improve the quality of the manuscript. I thus recommend the paper be reconsidered after minor revisions.

  1. It is suggested to change the title of the manuscript as follows

Current Development and Future Perspective on Natural Jute Fibers and their Biocomposites

  1. The introduction should be improved entirely so that the reader can identify the scientific progress of this review. For instance, there are several biocompatible and biodegradable biomaterials, including BSA, Gelatin, Zein, PCL, PLA, chitosan, UHMWPE, etc. but why did the authors select only Jute fibers/Jute fibers-based composites? The authors should highlight the favorable characteristics that made Jute fibers a more potent choice of biomaterials. Thus, the authors should emphasize why the Jute fibers are familiar, or favor compared to other biomaterials using the following literature. Moreover, the information on biomaterials (BSA, Gelatin, Zein, PCL, PLA, chitosan, UHMWPE, etc.) should be explored in the introduction with recent references or suitable sections, Thus, the following articles should be quoted in the introduction and other sections.

Chitosan (https://doi.org/10.1016/j.colsurfb.2021.111819), BSA (https://doi.org/10.1016/j.msec.2020.111698),

Gelatin; (https://doi.org/10.3390/ph14040291, https://doi.org/10.1016/j.jmbbm.2020.103696)

Zein (https://doi.org/10.3390/pharmaceutics11120621).

https://doi.org/10.1016/j.msec.2020.110928. https://doi.org/10.1039/D1EN00354B, https://doi.org/10.3390/pharmaceutics12121208

https://doi.org/10.1016/j.jmbbm.2021.104554,

It would be more realistic to cover such kind of research work in the current manuscript. Which will enrich the quality of the current manuscript as well as inquisitiveness to the readers.

  1. The mechanical properties of Jute fibers should be compared with the following forcspun fibers

https://doi.org/10.1039/D1EN00354B

  1. According to the corrections, the conclusions may be modified.

Reviewer 2 Report

Review: Manuscript Polymers 1647763

Title: Current Development on Natural Jute fiber and its Bio-Composites

The paper focuses on characterization of jute fibres and also presentation of research advancements in enhancing physical, mechanical, thermal, and tribological properties of the polymeric materials reinforced with jute fibres.

Abstract: The object of research, the aim of the work and major conclusions have been given in the abstract. The abstract has been written in accordance with the guidelines from the periodical.

The authors explained the need for the research in question while referring to relevant reference literature.

Relevant graphic documentation has been presented and the results have been discussed. In that part of the discussion, the mechanisms permitting the understanding of the test results have been explained.

Conclusion: The manuscript ends with the conclusion providing a summary of the activities undertaken in this work.

Reference: The reference literature consists of 146 items. The authors quote current reference literature relevant to the topic of the article. Quoting was done in accordance with the guidelines from the periodical.

Some shortcomings need to be corrected before publishing the article:

Fig. 2, correct the measurement unit for tensile strength (MPa)

Section 2.2. Jute Fiber/Yarn/Fabric Performance

2.2.1. Single jute fiber

What are the mechanical properties of jute fiber determined by [29]? They should be mentioned here.

In general, you might consider putting the comparisons between different researches into a table to help the reader quickly see what are the main results related the performance of jute fibres/fabric.
